# 4-Mercaptobenzoic Acid Labeled Gold-Silver-Alloy-Embedded Silica Nanoparticles as an Internal Standard Containing Nanostructures for Sensitive Quantitative Thiram Detection

**DOI:** 10.3390/ijms20194841

**Published:** 2019-09-29

**Authors:** Xuan-Hung Pham, Eunil Hahm, Kim-Hung Huynh, Byung Sung Son, Hyung-Mo Kim, Dae Hong Jeong, Bong-Hyun Jun

**Affiliations:** 1Department of Bioscience and Biotechnology, Konkuk University, Seoul 143-701, Korea; phamricky@gmail.com (X.-H.P.); greenice@konkuk.ac.kr (E.H.); huynhkimhung82@gmail.com (K.-H.H.); imsonbs@konkuk.ac.kr (B.S.S.); hmkim0109@konkuk.ac.kr (H.-M.K.); 2Department of Chemistry Education, Seoul National University, Seoul 151-742, Korea; jeongdh@snu.ac.kr

**Keywords:** ultrasensitive detection, thiram, internal standard, gold–silver-alloy-embedded silica nanoparticles

## Abstract

In this study, SiO_2_@Au@4-MBA@Ag (4-mercaptobenzoic acid labeled gold-silver-alloy-embedded silica nanoparticles) nanomaterials were investigated for the detection of thiram, a pesticide. First, the presence of Au@4-MBA@Ag alloys on the surface of SiO_2_ was confirmed by the broad bands of ultraviolet-visible spectra in the range of 320–800 nm. The effect of the 4-MBA (4-mercaptobenzoic acid) concentration on the Ag shell deposition and its intrinsic SERS (surface-enhanced Raman scattering) signal was also studied. Ag shells were well coated on SiO_2_@Au@4-MBA in the range of 1–1000 µM. The SERS intensity of thiram-incubated SiO_2_@Au@4-MBA@Ag achieved the highest value by incubation with 500 µL thiram for 30 min, and SERS was measured at 200 µg/mL SiO_2_@Au@4-MBA@Ag. Finally, the SERS intensity of thiram at 560 cm^−1^ increased proportionally with the increase in thiram concentration in the range of 240–2400 ppb, with a limit of detection (LOD) of 72 ppb.

## 1. Introduction

The use of pesticides in modern agriculture has improved crop yield and quality by controlling or destroying pests or weeds [1,2,3,4]. Although pesticides have diverse benefits, they are also a threat to consumer health because they are toxic to humans and other species [5,6]. When pesticides are used for crops or seeds, their traces could remain in the food [7], and these derivatives are considered to be toxic [8]. Further, pesticides are suspected to be carcinogenic and teratogenic compounds [7]. Therefore, the sensitive detection of a small concentration of these fungicides in soils, water, and foods, as well as their chemical state, is important [7].

Various methods have been proposed for monitoring pesticide residues, such as high-performance liquid chromatography (HPLC), gas chromatography-mass spectrometry (GC-MS), thin-layer chromatography, and enzyme-linked immunosorbent assay [9,10,11,12,13]. Currently, HPLC is the most robust and reliable method for food safety analysis. However, HPLC is time-consuming and expensive; requires a harsh solvent, high power source, bulky and sophisticated operation, complicated multi-step pre-treatment process; and could be dedicated in labs to trained personnel [14,15]. Thus, a fast, simple, highly sensitive, and stable method should be developed for the determination of pesticide residue.

Surface-enhanced Raman scattering (SERS) has been developed as a vibrational spectroscopy technique for various applications because of its non-destructive, rapid, molecular fingerprinting, ultrasensitive, and photostable properties [16,17,18,19,20]. Compared with HPLC-MS, SERS does not require harsh solvents and a high power source, and it is easily compatible with other detection systems [21]. As a result, many studies have focused on the use of different nanoparticles (NPs) as substrates for SERS detection of pesticides, such as silver nanostructures [21,22,23,24,25,26], gold nanostructures [27,28,29,30,31], and graphene oxide [32,33]. Although these nanostructures could enhance the SERS signal up to 10^14^ times, the practical application of SERS exhibits some technical challenges in the fabrication of reproducible, reliable, and robust SERS-active surfaces.

Recently, internal standards have been used to correct variations of SERS intensity in quantitative SERS assays [34,35,36,37]. Among them, the ratiometric SERS indicator-based detection mode of core-shell materials has been successfully developed because it can avoid the competition between the internal standard and the target molecules. However, difficulties in synthesizing an appropriate SERS probe for a specified target limited the application of the ratiometric SERS indicator-based detection mode [37]. Previously, our group reported Au-Ag alloys assembled silica NPs (SiO_2_@Au@Ag NPs) as a strong and reliable SERS probe with 4-mercaptobenzoic acid (4-MBA) as an internal standard located between the SiO_2_@Au core and the Ag shell. SiO_2_@Au@4-MBA@Ag NPs were synthesized by Au seed-mediated Ag growth on the surface of a silica template, followed by incorporating 4-MBA on the surfaces [38,39,40,41]. However, their application for SERS detection has not been completely investigated. In this study, we investigated the application of SiO_2_@Au@4-MBA@Ag NPs on pesticide detection.

## 2. Results and Discussion

To prepare SiO_2_@Au@4-MBA, silica NPs (ca. 150 nm in diameter) were first functionalized with amine groups by 3-aminopropyltriethoxysilane (APTS) to prepare aminated silica NPs, as shown in Figure 1 [42]. Simultaneously, colloidal Au NPs (3 nm) were prepared using tetrakis(hydroxymethyl)phosphonium chloride (THPC) and incubated with the aminated silica NPs by gentle shaking to prepare Au NPs embedded with SiO_2_ (SiO_2_@Au NPs), according to the method reported by Pham et al. [38,39,40,41]. Subsequently, 4-MBA was introduced on the surface of SiO_2_@Au NPs through the strong affinity between thiol groups and Au, and it was used as an internal standard. Finally, the Ag shell was deposited on SiO_2_@Au@4-MBA to enhance the Raman signal of RLCs by reducing a silver precursor (AgNO_3_) in the presence of ascorbic acid (AA) and polyvinylpyrrolidone (PVP) as a stabilizer and structure-directing agent under mild reducing conditions [39]. The silver ions reduced by AA were selectively grown onto SiO_2_@Au@4-MBA cores to form the core-shell SiO_2_@Au@4-MBA@Ag NPs; this was accompanied by an obvious color change to black. The presence of the Ag shell could also prevent the leakage of 4-MBA from the Au surface and improve the chance of generating numerous hot spots on the silica surface to detect the target molecules.

### 2.1. Characterizations of SiO_2_@Au@4-MBA@Ag NPs

We investigated the characteristics and effect of 4-MBA on the generation of SiO_2_@Au@4-MBA@Ag. Figure 2a shows the transmission electron microscopy (TEM) images of SiO_2_@Au@4-MBA@Ag. It can be seen that the Ag shell was well coated on its surface. The surface of SiO_2_ NPs was decorated with various small Ag NPs. From the TEM images, the average size of the SiO_2_@Au@4-MBA@Ag NPs was determined to be 195 ± 10 nm (*n* = 90). The zeta potential was also used to confirm the presence of Au NPs (Figure 2b). SiO_2_ NPs showed a zeta potential of −45 ± 0.1 mV. When the surface of the SiO_2_ NP was incubated with APTS, the zeta potential of SiO_2_@NH_2_ increased to −28 ± 0.6 mV because of the positive property of the NH_2_ groups. For all the NH_2_ groups, the Au NPs were immobilized on the surface of SiO_2_@NH_2_ by electrostatic attraction. The surface of the Au NPs was stabilized by THPC; therefore, the zeta potential of SiO_2_@Au decreased to −55 ± 6.1 mV. The sizes of SiO_2_@Au@4-MBA@Ag NPs increased when the Ag shell was deposited, as shown in Figure 2. The UV-Vis (ultraviolet–visible) spectra of SiO_2_@Au@4-MBA@Ag were consistent with the TEM images (Figure 2b). The suspension of SiO_2_ does not show its absorbance in the range of 300–1000 nm. Whereas the maximum peak of SiO_2_@Au was at ~520 nm when the Au NPs were immobilized on the surface of SiO_2_ NPs, the suspension of SiO_2_@Au@4-MBA@Ag NPs showed a broadband from 320 to 800 nm. This indicated the generation of irregular structures in the Ag shell and the creation of hot-spot structures on the surface of SiO_2_@Au@4-MBA@Ag NPs, producing a continuous spectrum of resonant multimode [38,39,40,41].

The Raman signals of SiO_2_@Au@4-MBA@Ag NPs were also measured (Figure 2d). The signal of 4-MBA on the surface of SiO_2_@Au NPs is unclear. In contrast, SiO_2_@Au@4-MBA@Ag exhibited a considerably stronger SERS signal of 4-MBA than SiO_2_@Au@4-MBA. In general, the bands of 4-MBA on the surface of SiO_2_@Au@4-MBA@Ag were observed at 360, 520, 715, 838, 1012, 1074, 1137, 1180, 1362, 1480, and 1582 cm^−1^ in Figure 2d. For 4-MBA, the peak at about 1074 cm^−1^ was attributed to the aromatic ring vibration possessing the C–S stretching mode, the band at about 1582 cm^−1^ arose from the aromatic ring breathing mode. The less intense band at 1362 cm^−1^ and 840 cm^−1^ were the COO^−^ stretching mode. Other weak bands at 1137 cm^−1^ and 1179 cm^−1^ corresponding to the C–H deformation modes were also observed. This result is consistent with our previous report [40,41,43]. The reproducibility and repeatability of Raman signals of SiO_2_@Au@4-MBA@Ag were showed in Appendix A. The sample was measured three times and repeated three times. The reproducibility and repeatability of Raman signals of SiO_2_@Au@4-MBA@Ag were calculated to be 2.7 and 8.1%, respectively. This result was rapidly similar to the size distribution of SiO_2_@Au@4-MBA@Ag by TEM analysis.

In addition, the effect of 4-MBA concentration on the SERS signal of SiO_2_@Au@4-MBA@Ag NPs was investigated. As previously reported, the density of carboxyl groups on the surface of SiO_2_@Au@4-MBA NPs affected the deposition of Ag shell on SiO_2_@Au@4-MBA [35,36]. Therefore, the effect of 4-MBA concentration on the SERS signal of SiO_2_@Au@4-MBA@Ag is considered in Figure 3. Various concentrations of 4-MBA in the range of 1–1000 µM were incubated with 100 µg of SiO_2_@Au@4-MBA, followed by Ag shell deposition of 300 µM AgNO_3_ in the presence of AA and PVP. All SiO_2_@Au@4-MBA@Ag NPs at 4-MBA in the range of 1–1000 µM were coated with Ag shells, as shown in Figure 3a. The Ag shell appears to have been better deposited at low concentrations than high concentrations of 4-MBA. The presence of irregular structures on the Ag shell on the surface of SiO_2_@Au@4-MBA@Ag NPs was also confirmed by UV–Vis spectroscopy with a broadband from 320 to 800 nm (Figure 3b). The SERS intensity of SiO_2_@Au@4-MBA@Ag NPs for 1–1000 µM 4-MBA clearly differed. The SERS signal of 4-MBA at all bands in Figure 3d increased gradually and became saturated after 100 µM. Therefore, we chose the concentration of 4-MBA concentration as 100 µM for further studies.

### 2.2. Detection of Thiram by SiO_2_@Au@4-MBA@Ag NPs

For the application, we chose thiram, a fungicide to prevent fungal diseases in seed and crops, as a pesticide sample in this study. Thiram is the simplest thiuram disulfide and the oxidized dimer of dimethyldithiocarbamate. In literature, the ratio of Raman intensity between a target molecule and an internal standard in quantitative SERS measurement provides more accurate information than the SERS signal of an intrinsic target molecule [31]. In our study, 4-MBA immobilized between the SiO_2_@Au core and Ag shell was used as an internal standard to calculate the concentration of thiram. Appendix A shows the SERS bands of SiO_2_@Au@4-MBA@Ag NPs in the presence and absence of thiram. Dominant bands of SiO_2_@Au@4-MBA@Ag were observed at 360, 520, 715, 838, 1012, 1074, 1137, 1180, 1362, 1480, and 1582cm^−1^. When thiram was adsorbed on the surface of SiO_2_@Au@4-MBA@Ag NPs, the SERS bands of thiram-incubated SiO_2_@Au@4-MBA@Ag NPs was observed at 360, 444, 520, 560, 715, 838, 881, 936, 1015, 1074, 1137, 1181, 1381, 1448, 1480, 151, 1582 cm^−1^. Thus, several new bands were obtained at 440, 560, 931, 1146, 1381, and 1512 cm^−1^. According to a previous report by Kang et al., SERS bands of thiram on the Ag surface was observed at 342, 446, 564, 870, 928, 1150, 1386, 1444, 1514 cm^−1^ [25]. Therefore, these bands were attributed to the characteristic bands of thiram [25,41]. The SERS bands of thiram and 4-MBA were partially overlapped; therefore, the SERS signals of thiram-incubated SiO_2_@Au@4-MBA@Ag NPs at 360, 881, 1074, 1137, 1181, 1381, 1582 cm^−1^ increased comparing to those of SiO_2_@Au@4-MBA@Ag NPs. To calculate the concentration of thiram, the ratio of Raman intensity of bands at 520 cm^−1^ and 560 cm^−1^ were chosen as the characteristic bands of 4-MBA and thiram, respectively.

#### 2.2.1. Optimization of Thiram Detection by SiO_2_@Au@4-MBA@Ag NPs

##### Effect of Employed Power Energy and Laser Lines

In the literature, the power energy has been considered an important factor affecting the SERS signal of target molecules. Therefore, we examined the effect of employed power energy on the SERS signal of thiram detection in the range of 2–10 mW (Appendix A). The SERS intensities of both SiO_2_@Au@4-MBA@Au in the presence or absence of thiram increased with the employed power energy. We chose the employed power of 10 mW to detect thiram for further studies. In order to investigate the effect of laser lines on the SERS signal of thiram, we also measured the Raman intensity of thiram-incubated SiO_2_@Au@4-MBA@Ag at the laser lines of 532 and 780 nm. As showed in Appendix A the Raman bands of thiram-incubated SiO_2_@Au@4-MBA@Ag using the laser line of 780 nm were broadened and unclear while those using the laser line of 532 nm was clearly obtained. Although UV-Vis absorbance might not be proportionally related to the strength of the SERS signal intensity, when that wavelength of the laser is irradiated to particle, the particles will absorb the laser energy well and it is likely to be connected to a strong SERS signal. Since we have 532 nm wavelength of the laser and the materials well absorbed the 300 nm to 650 nm wavelength with the maximum absorption peak at ~450 nm, we chose the laser line of 532 nm for Raman measurement.

##### Effect of Target Volume

Other than the power energy of the Raman equipment, the SERS signal was affected by various conditions of the nanomaterials or target concentration. We obtained the SERS signal of SiO_2_@Au@4-MBA@Au incubated with various volumes and concentrations of thiram; the results are shown in Figure 4a. The SERS signal at different concentrations of thiram (100, 500, and 1000 µL) was incubated with 20 µg of SiO_2_@Au@4-MBA@Ag. The ratio of SERS intensity between 4-MBA and thiram measured by the SERS signal of thiram in the range of 1–100 µM is shown in Figure 4. In Figure 4a, the SERS band ratios of thiram at 560 and 520 cm^−1^ were proportional to the thiram concentration at all thiram volumes. However, the behavior of thiram for each specific volume varied with the concentration. At 100 µL thiram, the SERS band of thiram began increasing at 5 µM, while this ratio increased immediately at 1 µM when 500 µL and 1000 µL thiram were incubated. In addition, the SERS signal of thiram was almost balanced very early at 5 µM, when 1000 µL thiram was utilized. This is because the quantities of thiram at each volume changed with the thiram concentration, while the surface area of SiO_2_@Au@4-MBA@Ag was constant (20 µg). We found that when 100 µL of 5 µM thiram or 500 µL of 1 µM thiram was utilized, the quantity of thiram was calculated to be 0.5 nmol, and the SERS signal of thiram can be observed at 0.5 nm. The SERS signal of thiram was balanced at 5 nmol. The SERS signal of thiram increased steadily and slowly at 500 µM thiram; hence, we chose 500 µL of thiram for further studies.

##### Effect of Quantity of SiO_2_@Au@4-MBA@Ag NPs

The effect of SiO_2_@Au@4-MBA@Ag quantity is also considered in Figure 4. SiO_2_@Au@4-MBA@Ag amounts of 10, 20, and 30 µg were incubated with 500 µL thiram at 1 to 100 µM thiram, and the results are shown in Figure 4b. For the same concentration of thiram, the greater the amount of SiO_2_@Au@4-MBA@Ag added to the thiram solution acting as a substrate for thiram detection, the lower the SERS signal of thiram (Figure 4b). It is well known that the efficiency of SERS depends on the density of the target on the surface of nanomaterials [19]. Therefore, when a greater amount of SiO_2_@Au@4-MBA@Ag was added, more hot spots were available on the surface of SiO_2_@Au@4-MBA@Ag, generating numerous detection sites for thiram, meanwhile, the quantity of thiram was constant. Thus, the density of thiram at the gap or on the surface of SiO_2_@Au@4-MBA@Ag decreased, resulting in decreased enhancement of the thiram intensity between two adjacent Au@4-MBA@Ag NPs on the surface of SiO_2_@Au@4-MBA@Ag (Figure 4c) [33]. As a result, the intensity of thiram-incubated SiO_2_@Au@4-MBA NPs decreased (Figure 4b). For the same concentration of thiram, the greater the amount of SiO_2_@Au@4-MBA added, the larger the gap between Au@4-MBA@Ag NPs (Figure 4a).

##### Effect of Incubation Time of Thiram

Incubation is an important factor that affects the adsorption of target molecules onto the surface of a nanomaterial. Thus, the effect of thiram incubation time on the SERS signal is shown in Figure 4c. The SERS intensity of thiram-incubated SiO_2_@Au@4-MBA@Ag was proportional to the incubation time, and the highest value was achieved at 30 min. The result indicated that thiram was absorbed efficiently on the surface of SiO_2_@Au@4-MBA@Ag because of the thiol groups.

##### Effect of Concentration of Thiram Incubated SiO_2_@Au@4-MBA@Ag

According to a previous report, the density of nanomaterials significantly affected the SERS signal. Figure 4d shows the effect of thiram-incubated SiO_2_@Au@4-MBA@Ag concentration on the SERS signal of thiram. In the absence of thiram, the SERS signal of SiO_2_@Au@4-MBA@Ag NP suspension was insignificantly different; meanwhile, in the presence of thiram, the SERS signal of SiO_2_@Au@4-MBA@Ag NPs increased slowly from 50 µg/mL to 100 µg/mL and achieved the highest value at 200 µg/mL. For concentrations greater than 200 µg/mL, the SERS signal of the thiram-incubated SiO_2_@Au@4-MBA@Ag NP suspension decreased sharply owing to the low diffraction of the suspension.

#### 2.2.2. Detection of Thiram by SiO_2_@Au@4-MBA@Ag NPs

We measured SERS signals at various concentrations of thiram, in the range of 240–24,000 ppb, with 20 µg of the SiO_2_@Au@4-MBA@Ag. The bands at 520 cm^−1^ and 560 cm^−1^ were chosen as the characteristic bands of 4-MBA and thiram, respectively. The SERS intensity of thiram at 560 cm^−1^ and the ratios of the Raman intensity of thiram to that of 4-MBA are shown in Figure 5. In Figure 5a, the SERS intensity at 560 cm^−1^ increased with the increase in a thiram concentration lower than 2400 ppb. Whereas the SERS intensity ratio at 560 cm^−1^ and 520 cm^−1^ increased proportionally with the increase in thiram concentration lower than 12,000 ppb (Figure 5b). The calibration curves of thiram showed a linear dependence (y = 0.344 × C + 6.625; *R^2^* = 0.95 (C = thiram concentration, y is SERS signal value on y-axis in Figure 5c)) between the SERS intensity ratio and thiram concentration between 240 ppb and 2400 ppb. The limit of detection of thiram was determined to be 72 ppb (S/N (signal to noise ratio) = 3), which is considerably lower than 288.5 ppb, as in our report [35], and the limit of the detection value is lower than the maximal residue limit recommended by the US (7 ppm) and Canada (0.1 ppm) [44,45].

## 3. Materials and Methods

### 3.1. Chemicals and Reagents

Tetraethylorthosilicate (TEOS), APTS, silver nitrate (AgNO_3_), chloroauric acid (HAuCl_4_), THPC, 4-mercaptobenzoic acid (4-MBA), AA, PVP, and thiram were purchased from Sigma-Aldrich (St. Louis, MO, USA) and used without further purification. Ethyl alcohol (EtOH) and aqueous ammonium hydroxide (NH_4_OH, 27%) were purchased from Daejung (Siheung, Korea). Ultrapure water (18.2 MΩ cm) was produced using a Millipore water purification system (EXL Water purification, Vivagen Co., Ltd., Seongnam, South Korea). Thiram: the toxicity class WHO III (LD_50_ for rabbits >210 mg/kg, Inhalation LC_50_ (4 h) for rats 4.42 mg/kg) [46].

### 3.2. Preparation of SiO_2_@Au@4-MBA

In a previous report, Pham et al. revealed that the SiO_2_@Au@Ag NPs possessed a relatively high Raman enhancement effect [38,39,40,41]. Au NPs assembled silica nanoparticles (SiO2@Au NPs) were prepared by incubating the Au NP suspension with aminated silica NPs overnight. Subsequently, 1 mL of 100 µM 4-MBA solution in EtOH was added to SiO_2_@Au (1.0 mg), and the suspension was stirred vigorously for 1 h at room temperature. The colloids were centrifuged and washed several times with EtOH. The NPs were re-dispersed in 1.0 mL absolute EtOH to obtain 1 mg/mL SiO_2_@Au NPs modified with 4-MBA (SiO_2_@Au@4-MBA).

### 3.3. Preparation of SiO_2_@Au@4-MBA@Ag NPs

Au@4-MBA@Ag NPs assembled silica NPs were prepared in an aqueous medium via the reduction and deposition of Ag using ascorbic acid onto SiO_2_@Au@4-MBA NPs in a polyvinylpyrrolidone (PVP) environment. Briefly, 200 µL of 200 µg/µL SiO_2_@Au@4-MBA was dispersed in 9.8 mL of water containing 10 mg PVP, which was kept still for 30 min. Then, 20 µL of 10 mM silver nitrate was added to the suspension, followed by the addition of 20 µL of 10 mM ascorbic acid. This suspension was incubated for 15 min to completely reduce the Ag^+^ ions to Ag. The reduction steps were repeated to obtain the desired AgNO_3_ concentration. SiO_2_@Au@4-MBA@Ag NPs were obtained by centrifuging the suspension at 8500 rpm for 15 min, and the NPs were washed several times with EtOH to remove excess reagent. SiO_2_@Au@4-MBA@Ag NPs were re-dispersed in 1 mL of absolute EtOH to obtain a 200 µg/mL SiO_2_@Au@4-MBA@Ag NP suspension.

### 3.4. Thiram Detection

To absorb thiram on the surface of SiO_2_@Au@4-MBA NPs, 500 µL of 50 µM thiram solution was incubated with 500 µL of 200 µg/mL SiO_2_@Au@4-MBA@Ag NPs suspension for 30 min, followed by centrifugation for 15 min at 13,000 rpm to collect the colloids. The prepared NPs were washed several times with EtOH to remove excess reagent. SiO_2_@Au@4-MBA@Ag@thiram NPs was re-dispersed in 500 µL of absolute EtOH to obtain a 200 µg/mL SiO_2_@Au@4-MBA@Ag@thiram NP suspension.

### 3.5. SERS Measurement of SiO_2_@Au@4-MBA@Ag@thiram

To obtain the surface-enhanced Raman spectrum, the obtained colloids suspensions were measured in a capillary tube. SERS signals were measured using a micro-Raman system with a 532 nm laser excitation source and equipped with an optical microscope (BX41, Olympus, Tokyo, Japan). The SERS signals were collected in a back-scattering geometry using a ×10 objective lens (0.90 NA, Olympus, Shinjuku, Tokyo, Japan). A 532 nm diode-pumped solid-state laser was used as a photo-excitation source, exerting a laser power of 10 mW at the sample. The selected sites were measured randomly, and all SERS spectra were integrated for 5 s. The size of the laser beam spot was approximately 2.0 μm. The SERS spectrum was obtained in the 300–2000 cm^−1^ wavenumber range.

## 4. Conclusions

We successfully prepared SiO_2_@Au@4-MBA@Ag nanomaterials and optimized their conditions for thiram detection. The presence of Au@4-MBA@Ag alloys on the surface of SiO_2_ was confirmed by the broad bands in the range of 320 to 800 nm, indicating the generation of bumpy structures on the the Ag shell. The effect of 4-MBA concentration on the SERS signal of SiO_2_@Au@4-MBA@Ag NPs was studied. The SERS signal of 4-MBA increased gradually for concentrations under 100 µM. For thiram detection, SiO_2_@Au@4-MBA@Ag exhibited a stronger SERS signal of 4-MBA at 360, 520, 715, 838, 1012, 1074, 1137, 1362, 1480, and 1582 cm^−1^. Meanwhile, several new bands of thiram were obtained at 440, 560, 931, 1146, 1381, and 1512 cm^−1^ when thiram was adsorbed on the surface of SiO_2_@Au@4-MBA@Ag NPs. Additionally, the SERS intensities of both SiO_2_@Au@4-MBA@Au increased with the employed power energy from 2 to 10 mW. The SERS intensity of the thiram incubated SiO_2_@Au@4-MBA@Ag achieved the highest value via incubation with 500 µL thiram for 30 min and measuring SERS at 200 µg/mL SiO_2_@Au@4-MBA@Ag. Finally, the SERS intensity of the thiram at 560 cm^−1^ increased proportionally with the increase in thiram concentration in the range of 240 to 2400 ppb with a LOD of 72 ppb. This study provides a thorough understanding of thiram detection, which supports further research and development for strong and reliable SERS probes based on SiO_2_@Au@4-MBA@Ag NPs.

## Figures and Tables

**Figure 1 ijms-20-04841-f001:**
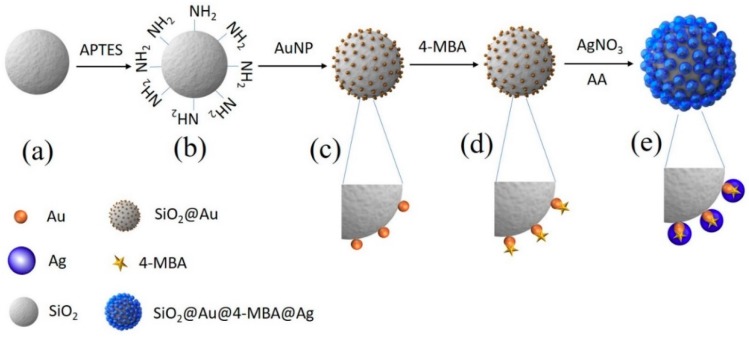
Illustration of preparation of Au@4-MBA@Ag embedded silica nanoparticles (SiO_2_@Au@4-MBA@Ag NPs) for surface-enhanced Raman scattering probe. (**a**) Silica NP, (**b**) aminated silica NP, (**c**) Au NPs embedded silica NP, (**d**) Au NPs embedded silica NP incubated with 4-MBA (SiO_2_@Au@4-MBA) and (**e**) SiO_2_@Au@4-MBA coated with Ag shell by the reduction of silver nitrate in the presence of ascorbic acid and polyvinyl pyrrolidone.

**Figure 2 ijms-20-04841-f002:**
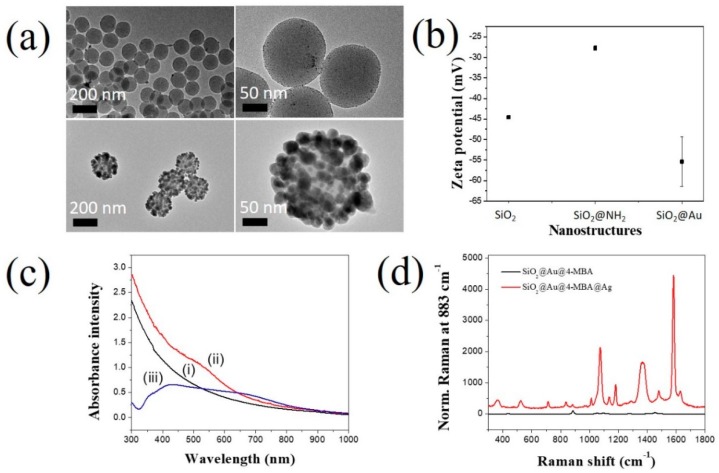
Characteristics of SiO_2_@Au@4-MBA@Ag. (**a**) TEM images, (**b**) Zeta potential, (**c**) UV-Vis spectra of (i) 1000 µg/mL SiO_2_, (ii) 1000 µg/mL SiO_2_@Au@4-MBA, and (iii) 10 µg/mL SiO_2_@Au@4-MBA@Ag, and (**d**) Raman spectra of SiO_2_@Au@4-MBA and SiO_2_@Au@4-MBA@Ag. Error bar represents the average value of three samples.

**Figure 3 ijms-20-04841-f003:**
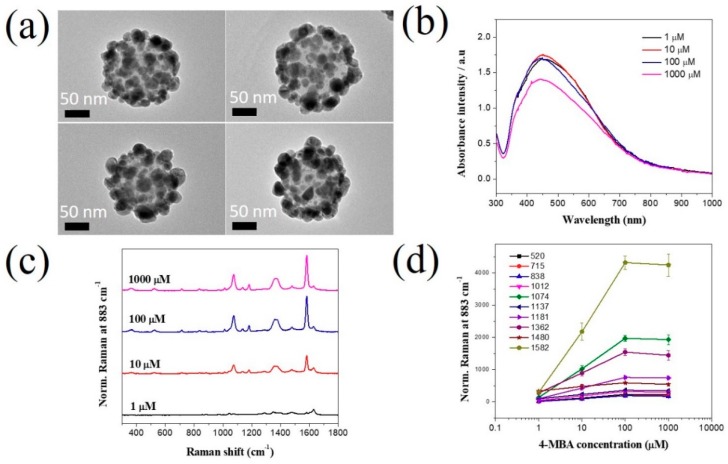
Effect of 4-MBA on SERS signal of SiO_2_@Au@4-MBA@Ag NPs at different concentrations in the range of 1 µM–1000 µM. (**a**) TEM images, (**b**) UV-Vis spectra, (**c**) Raman spectra, and (**d**) Raman signal plot of SiO_2_@Au@4-MBA@Ag. Error bar represents samples in triplicate.

**Figure 4 ijms-20-04841-f004:**
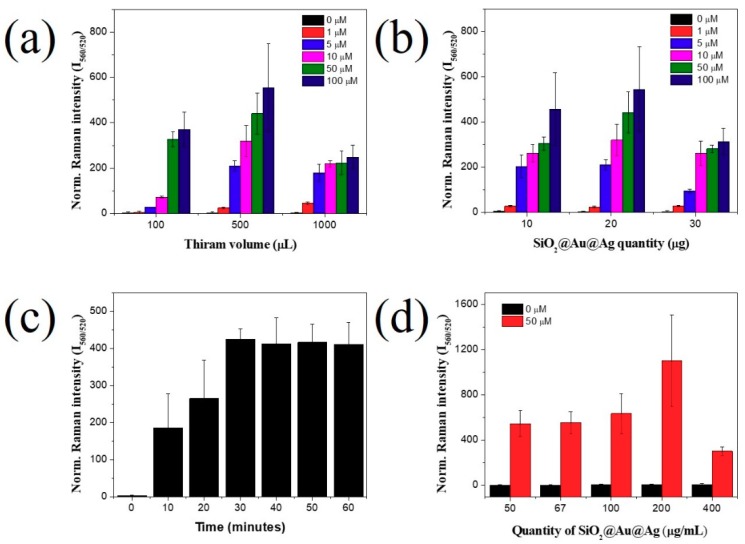
Effect of (**a**) thiram volume, (**b**) quantity of SiO_2_@Au@4-MBA@Ag, (**c**) incubation time, and (**d**) dilution of SiO_2_@Au@4-MBA@Ag nanoparticles in the presence of 50 uM thiram. Error bar represents the triplicates of samples.

**Figure 5 ijms-20-04841-f005:**
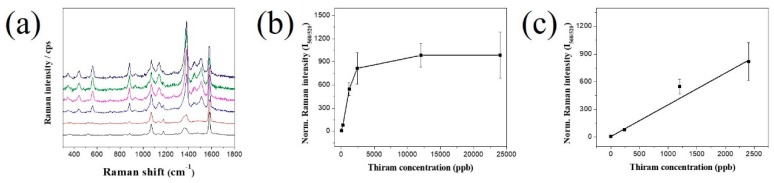
(**a**) Raman signal (different lines from bottom to top represents different thiram concentration: 0, 240, 1200, 2400, 12,000 and 24,000 ppb) and (**b**) Raman intensity ratio at 560 and 520 cm^−1^ and (**c**) calibration curves of thiram-incubated SiO_2_@Au@4-MBA@Ag NPs at various concentrations of thiram from 0 to 24,000 ppb. The dynamic linear range of 240 to 2400 ppb with the limit of detection of 72 ppb. Error bar represents an average value of five samples.

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
