# Peer review of "4-Mercaptobenzoic Acid Labeled Gold-Silver-Alloy-Embedded Silica Nanoparticles as an Internal Standard Containing Nanostructures for Sensitive Quantitative Thiram Detection"

_ijms, 2019, doi:10.3390/ijms20194841_

Round 1

Reviewer 1 Report

The authors report on a nanoplasmonic composite material for thiram detection via SERS analysis. The manuscript is well written, the experiments are well designed and the approach seems to be technically sound. However, the points below should be addressed prior to publication:  

In order to specify the context of the manuscript, the abstract should specify that thiram is a pesticide. Suspensions of the studied materials should be displayed, for example in Figure 1. Generally, particles are dispersed as suspensions not as solutions. The authors should correct this term throughout the manuscript to avoid misconceptions. Figure 2. UV-Vis and Raman spectra of the explored SiO2 particles should also be included. Why did the authors choose a 532 nm laser and no other laser line/s? Please, elaborate. The authors are expected to include experiments with other laser lines to justify this. The intensity of the power source is not enough to describe the employed power. In order to support repeatability and reproducibility of the described method, the authors are expected to include the employed size of the spot (of the laser during Raman analysis). The particle size distribution of the resulting nanoparticles should be carefully characterized and discussed. The Raman signatures resulting throughout the analytical process should be displayed in Figure 5. Regulation of thiram content in environmental samples should be discussed. Is the analytical range of the described method meeting such regulations? Figure 2-5. What do the error bars represent, respectively? Please clarify in the corresponding caption. Risks / Safety issues related to the experimental section are not described, which should be carefully included.

Reviewer 2 Report

//

ijms-582788

The manuscript entitled “Ultrasensitive detection of thiram using 4-mercaptobenzoacic labeled gold–silver-alloy-embedded silica nanoparticles as an internal standard containing nanostructures” by Xuan-Hung Pham, Eunil Hahm, Kim-Hung Huynh, Byung Sung Son, Hyung-Mo Kim, Dae Hong Jeong, and Bong-Hyun Jun reports on preparation of Au–Ag-alloy-embedded SiO2 NPs which were used for thiram detection.

The manuscript has potential but needs some more work before it can be accepted for publication. I am concerned about few factors of the study as listed below:

1) The title seems to be corrected in order to be much clear.

2) The biggest portion of the paper, which describes preparation of SiO2@Au@4-MBA, silica NPs and their characterization are adequate. Unfortunately I have some serious concerns about SERS study of the fungicide thiram.

Thiram belongs to the dithiocarbamate (DTC) pesticides. Some other DTC are thiram, ferbam, ziram, metiram, zineb and mancozeb. They are widely studied. See for example refs: Vibrational Spectroscopy, 17 (1998) 133; Bull. Korean Chem. Soc., 23 (2002) 1604; Langmuir, 25 (2009) 13833; Analyst, 137 (2012) 5082; Applied Spectroscopy, 73 (2019) 313. I think they can be cites regardless difference of the active substrates or solutions used.

3) Can the authors comment on fig. S1. What I understand from it is the increase of the intensities of the peaks of the SERS spectrum of SiO2@Au@4-MBA@Ag NPs when 50 μM thiram is introduced. Here peaks of thiram, which are quite shallow, are not indicated by the authors but are given in the manuscript (lines 133-138).

How the authors calculated concentration of thiram? In fact they calculated the ratio between peaks of thiram with those of the NPs spectrum. Thus, they compared different bands, assigned to two different chemicals, i.e. thyram (560 cm−1) and SiO2@Au@4-MBA@Ag NPs (520 cm-1). I think this is an wrong procedure. Two peaks have different origin and the authors have used them for calculation?

This I identify as major deficiency in the manuscript implying also a limited impact that the manuscript would have in its current form.

4) No assignment of the band is provided.

5) Fig. 2 d – no “y” units; fig. 3 c, d - no “y” units; fig. 4 a, b, d – delete dimensions from the scales; fig. 4 a, b, c, d - no “y” units; fig. 5 a, b - no “y” units;

6) Lastly, there are some spelling errors which need to be corrected.

I am emphasizing that only a very substantial improvement can lead to the manuscript becoming publishable.

Round 2

Reviewer 1 Report

The authors revised the manuscript according to the comments and suggestions. It can be accepted for publication as it is.